# Reproductive Dynamics of the Seabob Shrimp *Xiphopenaeus kroyeri* in Trawl Fisheries in Southeastern Brazil

**DOI:** 10.3390/biology14070758

**Published:** 2025-06-25

**Authors:** Amanda Soares dos Santos, Cecília Fernanda Farias Craveiro, Hildemário Castro-Neto, Caroline Vettorazzi Bernabé, Douglas da Cruz Mattos, Leonardo Demier Cardoso, Adriano Teixeira de Oliveira, Paulo Henrique Rocha Aride, Henrique David Lavander, Maria Aparecida da Silva

**Affiliations:** 1Postgraduate Program in Veterinary Sciences, Federal University of Espírito Santo, Alegre 29500-000, Brazil; amandasoaresds@gmail.com (A.S.d.S.); mvmariaaparecida@gmail.com (M.A.d.S.); 2Department of Fisheries and Aquaculture, Federal Rural University of Pernambuco, Recife 52171-900, Brazil; ceciliacraveiro@yahoo.com.br (C.F.F.C.); hildemariocastro@gmail.com (H.C.-N.); 3Laboratory of Nutrition and Production of Aquatic Organisms, Federal Institute of Espírito Santo, Piúma 29285-000, Brazil; vettorazzicarol@gmail.com (C.V.B.); douglas_uenf@yahoo.com.br (D.d.C.M.); leonardodemier@hotmail.com (L.D.C.); henrique.lavander@ifes.edu.br (H.D.L.); 4Center for Studies of Invertebrates and Vertebrates of the Amazon (NEIVA), Federal Institute of Amazonas, Manaus 69083-475, Brazil; aride@ifam.edu.br

**Keywords:** closed season, fisheries impact, gonadal development, maturation, Penaeidae, reproduction, reproductive biology, seabob shrimp

## Abstract

In the western Atlantic, seabob shrimp *Xiphopenaeus kroyeri* is subject to significant capture pressure, necessitating periods of fishing stoppage to safeguard the species’ wild propagation. The following five stages of maturation were noted: immature, beginning, advanced, mature, and spawned. The average size of the female cephalothorax at the first gonadal maturation was 1.7 cm, and the maximum proportion of mature females was seen in May and July. There were more juveniles during the recruitment phase in April. The number of adult females was negatively correlated with precipitation. The life cycle of the *X. kroyeri* population on the Espírito Santo coast of Brazil occurs essentially between the months of April, May, and July.

## 1. Introduction

Crustaceans are the second most common group of aquatic organisms used as a food source worldwide. The species *X. kroyeri* (Heller, 1862) is found in the western Atlantic, from North Carolina in the United States to the State of Rio Grande do Sul in Brazil [1,2]. It is a species that inhabits coastal and shallow oceanic waters, living on sandy–muddy bottoms at depths of up to 30 m, and is traditionally caught by bottom trawling [3]. Recently, it was reported for the first time as an invasive species in Mediterranean waters. It was the first shrimp species endemic to the western Atlantic to be introduced and establish itself in the Mediterranean Sea, becoming part of the commercial shrimp fishery in Egypt [4].

The largest shrimp productions globally are achieved through trawl fishing; however, this method does not show selectivity. Once captured alongside the target species, many bycatch fauna are also captured during trawling. This technique causes a significant effect on the sea floor and generates too much bycatch in fishing. Thus, an effective fishery management requires well-defined strategies, such as the closed period, which helps in the preservation of fish stocks during their reproduction/recruitment period, and incentives for testing and using devices to reduce bycatch fauna (BRDs—Bycatch Reduction Devices) in shrimp fishing [3,5,6].

In addition to the recognized economic potential generated by fishing, the species plays a highly relevant biological role, since it occupies a position at the trophic base, transferring energy from primary productivity to the upper food chain. The biological importance of this resource reinforces the need for sustainable management of the species in order to reduce the risk of harmful cascading effects which may affect the productivity of commercially exploited fish [7]. Understanding the dynamics of a stock is essential for assessing the compatibility between mating and sexual maturity, as well as the timing of these events for a given species. This is essential for formulating appropriate management practices in fishery science, allowing for adequate management of an exploited population and helping preserve the species [8,9,10].

Knowing the reproductive phases of a species is essential to understand the reproductive dynamics of fishing stocks. Macroscopic analysis is the most commonly used method for describing and classifying gonadal development in shrimp, based mainly on observing the gonads’ color pattern. The mature specimens’ colors vary from transparent to yellowish, orange-yellow, olive-brown, and green. They also vary in size, which allows the identification of the stage of development with a macroscopic examination.

Classifying ovarian maturation stages using the color scale is practical for maturation management [11,12]. Histological analysis is the most recommended method for achieving high accuracy [13,14]. Several authors have carried out studies addressing the reproductive dynamics of the species *X. kroyeri* along the Brazilian coast [15,16,17,18,19,20,21,22]. However, the association between macroscopic and histological analyses as a tool for studying the reproductive dynamics of *X. kroyeri* is still scarce. Thus, the present study aimed to describe the stages of gonadal development and the reproductive dynamics of the seabob shrimp *X. kroyeri* captured on the southern coast of the state of Espírito Santo to provide information for the elaboration of recommendations for the sustainable fishing of this species, which has great biological and economic importance.

## 2. Materials and Methods

### 2.1. Sampling

Shrimp were captured from March 2019 to February 2020 in the coast of the municipalities of Anchieta (20°48.547′ S, 40°38.309′ W), Piúma (20°50.609′ S, 40°44.485′ W), and Itapemirim (21°0.974′ S, 40°48.827′ W), on the southern coast of Espírito Santo, in the southeast region of Brazil (Figure 1) (SISBIO no. 67056-1). A motorized fishing boat was used. It operated with a 12 m long double trawl net, 2.5 m wide at the mouth, 30 mm internode distance in the sleeves, and a net body with 15 mm internode distance in the bagger.

A monthly shipment was carried out, with two hauls lasting 60 min each, for 24 hauls in the 12 months of analysis. Each shipment was considered a sampling unit, and an artisanal fisherman aided all of them. The geographic coordinates were recorded with a Sonar GPS (GARMIN^®^-Echomap 54CV, Olathe, KS, USA) at the beginning and end of each trawl.

After carrying out the trawl, the captured shrimp were separated from the accompanying fauna on the vessel’s deck. They were then packed in plastic bags, stored in an ice cooler, and taken to the Laboratory of the Federal Institute of Espírito Santo—Campus Piúma, Espírito Santo, Brazil. In the laboratory, the fresh shrimp were identified according to the species, based on the identification guide for penaeid shrimp [23]. After identification, the specimens were separated by sex based on external characteristics. Females were identified by observing the thelycum, while males were identified by observing the petasma and terminal ampullae.

All specimens were measured for total length (TL) and cephalothorax length (CL) using a caliper (0.05 mm). The wet weight (W) was analyzed using a precision analytical scale (0.001 g).

The Kolmogorov and homoscedasticity tests were applied to assess the data’s normality. The sex ratio for all sampling months was estimated using the Chi-square test with a 5% significance level (*p* < 0.05). For statistics, the non-parametric Wilcoxon test was used to compare medians when data were not normal (Shapiro–Wilk and Levene), with a 5% significance level (*p* < 0.05). Regression analyses were performed using the least squares method, with the Equation y = a + bx for linear regression, and y = a + x^b^ for non-linear regression.

### 2.2. Macroscopic Analysis of Males and Females

An average of 300 male and female shrimps were randomly sampled monthly. The females were classified according to the macroscopic maturational stage based on the observation of the morphology of the ovaries through the shrimp carapace, as well as the coloration of the ovaries using a widely available color scale (Pantone Matching System^®^, Coated Simulation, Pantone, Carlstadt, NJ, USA), following the macroscopic characteristics proposed by Bernabé et al. [24] for this species. The male’s maturation status was determined by the degree of union of the petasma; individuals with united petasma were considered mature, and those with disunited petasma were considered immature [25].

### 2.3. Microscopic Analysis of Ovarian Development

Histological analyses of the female gonads were performed with a monthly subsample of 20 females (*n* = 240), chosen randomly from the total number of females macroscopically sampled. The sample size was based on studies by Souza et al. [11]. After macroscopic analysis of the color, the gonads were weighed and fixed in Davidson’s solution for 24 h and transferred and preserved in 70% alcohol.

Fragments of 1–2 mm from the median portion of the gonads were dehydrated, cleared in xylene, and finally impregnated and embedded in paraffin at 65 °C. Then, the samples were sectioned (5 µm) using a LEICA RM2125RT, Wetzlar, Germany, rotary microtome and stained with Hematoxylin/Eosin-Floxin, according to the methodology of Junqueira and Junqueira [26]. The sections were then examined under a LEICA DM500 optical microscope with a LEICA ICC50HD camera attached. They were photographed to characterize the maturational stage and measure the oocytes. As for the classification of oocytes and their respective stages of development, the histological characteristics proposed by Craveiro et al. [27] were adapted for the species.

Three hundred oocytes were randomly measured with sectioned nuclei at maximum diameter in all maturational stages. The ImageJ software version 1.53k for Windows (Wayne Rasband and contributors, National Institutes of Health, Miami, FL, USA) was used for measuring. The measured data were gathered according to the presence of the most developed cell stage in each maturation stage.

Size at first maturation was estimated using the equation described by King [28] P = 1/(1 + exp [−r (L − Lm)]), where P is the percentage of adult females in a length class, r is the slope of the curve, L is the upper limit of the length class, and Lm is the average length at first maturity, using the sizeMat package of the R software. The ovaries were classified into five stages of maturation (immature = I, in initial maturation = IIM, in advanced maturation = IAM, mature = M, and spawned = S) and for the model, the cephalothorax lengths of adult animals were used (only the IIM, IAM, M, and S stages).

Data referring to the rainfall index (RI) (monthly for the previous 10 years) and Sea Surface Temperature (SST) (weekly averages) were obtained from the Capixaba Institute of Research, Technical Assistance and Rural Extension (Incaper) and the National Institute for Space Research (INPE), with the Center for Weather Forecasting and Climate Studies (CPTEC), respectively. The Kolmogorov and homoscedasticity tests were applied to assess the data’s normality. Thus, the non-parametric Spearman and rank test was used to determine the association between mature females, rainfall, and temperature, with a 5% significance level of 5% (*p* < 0.05). Statistical analyses were performed using the R software (R-3.4.4 for Windows).

## 3. Results

In total, 3658 specimens of *X. kroyeri* were captured during the 12 months of sampling. Of these, 1831 specimens were females (50.05%) and 1827 were males (49.95%). No significant difference was found (*p* > 0.05) in the sex ratio of females and males collected, which is equivalent to 1:1 (female–male). The months that showed a significant difference were April, with a higher proportion of females, and October, with a higher proportion of males (*p* < 0.05).

Adult females had a mean ± standard deviation of 10.30 ± 1.36 cm (11.00–9.10 cm) in total length and 2.13 ± 0.33 cm (2.31–1.92 cm) in cephalothorax length. For young females, the data were 7.27 ± 1.46 cm (7.51–6.50 cm) in total length and 1.49 ± 0.31 cm (1.67–1.30 cm) in cephalothorax length.

Adult males had a mean ± standard deviation of 9.25 ± 1.16 cm in total length and 1.83 ± 0.27 cm in cephalothorax length. Young males had 7.21 ± 1.35 cm in total length and 1.44 ± 0.30 cm in cephalothorax length.

The wet weight showed a minimum of 0.6 g and a maximum of 18.2 g, with an average of 6.21 ± 2.47 g for adult females. For young females, the minimum wet weight was 0.4 g and the maximum was 11.3 g, averaging 2.27 ± 1.51 g. The damp weight for adult males showed a minimum of 0.4 g and a maximum of 12.2 g, with an average of 4.44 ± 1.64 g. For young males, the minimum wet weight was 0.5 g, the maximum was 8.5 g, and the average was 2.18 ± 1.44 g.

Statistical analyses showed a significant difference between the variables of total length, length of the cephalothorax, and weight between males and females of *X. kroyeri* captured in the southern coast of Espírito Santo, in which females are larger and heavier than males (*p* < 0.05).

Males had a higher number of individuals in the cephalothorax length of from 1.5 to 2.0 cm (Figure 2). Compared to males, we observed a predominance of females below 1.5 cm and above 2.1 cm. A greater number of female individuals up to 3.5 cm was also observed, confirming that females are longer than males.

Five maturational stages—I, IIM, AAM, M, and S—were verified in the macroscopic and histological analysis of the ovaries. The immature stage (I) histologically evidenced the primordial oogonia oocytes (OO) measuring 6.51 ± 2.32 µm (7.31–5.21 µm) in diameter and pre-vitellogenic oocytes (PVTO) measuring 26.77 ± 11.95 µm (35.31–19.21 µm) (Figure 3A). In the initial maturation (IIM), oocytes in early vitellogenesis (OEV) measuring 90.02 ± 16.60 µm (75.01–99.89 µm) were evidenced (Figure 3B). Advanced maturation (IAM) revealed oocytes in advanced vitellogenesis (OAV) measuring 123.31 ± 26.95 µm (115.01–99.92 µm) (Figure 3C). The mature stage (M) showed mature oocytes (MO) measuring 146.51 ± 21.79 µm (159.74–134.01 µm) (Figure 3D). After spawning, atretic oocytes (AO) were seen (Figure 3E).

The gonadosomatic index (GSI%) relates the representativeness of the gonad weight to the total shrimp weight and presents the highest averages in September and April. Table 1 shows the gonad weight, gonadosomatic index (mean ± standard deviation), and the statistical differences for each maturational stage.

Regarding the distribution of gonadal maturation stages during the sampling months (Figure 4), a higher percentage of mature females was observed in May and July. Females in advanced maturation were abundant in October, December, January, and February. Spawned females were more abundant in June, August, and September, but mature and spawned females were present in all months.

The average size of the cephalothorax at the first gonadal maturation CL50 of females (Figure 5) was estimated to be 1.7 cm, with R^2^ = 0.8 indicating a good fit of the first maturation curve. For the total length parameter TL50, the value corresponded to 8.4 cm. In April (Figure 6), there was the highest rate of juvenile individuals, indicating the recruitment period for the species in this region.

The water temperature ranged from 22.7 to 26.7 °C during the sampling months of this study. The highest and lowest temperature peaks were observed in April and August. The highest average monthly rainfall index of the last 10 years was in November, and the lowest was in February. The Spearman rank correlation showed significant associations between the percentage of mature females and rainfall index (RI) (*p* < 0.05), but not for the ratio of mature females to sea surface temperature (SST) (*p* > 0.05). The increase in rainfall is directly related to the reduction in mature females in the period, as observed in March and November (Figure 7).

## 4. Discussion

The sex ratio of 1:1 (female–male) found in this study corroborates the research reported by Eutropio et al. [16] carried out in this municipality of Anchieta, Espírito Santo, by Castilho et al. [19] carried out in São Paulo, and by Reis JR et al. [20] and Silva et al. [21], both carried out in Sergipe. According to Coelho and Santos [29], when this proportion of males and females is approximately equal, it may indicate that the explored collection area may be a possible mating place for the species. Thus, the region studied here is a mating place for *X. kroyeri*. Lopes et al. [18], with the same species in Pernambuco, showed a different sex ratio than the one found here, equivalent to 1:0.78 (female–male).

The ratio of total length, cephalothorax length, and total weight of specimens showed that females of *X. kroyeri* are significantly larger and heavier than males. Studies by Eutropio et al. [16], Martins et al. [17], Reis JR et al. [20], and Silva et al. [21] on the same species also demonstrate higher length and weight patterns in females compared to males. According to Dall et al. [23], the differences in the parameters of size and weight, where females exhibit greater amplitude compared to males, can be explained by the fact that females need space in the cephalothoracic cavity, where the gonads are located; thus, this greater size is due to their greater development.

In this study, five maturation stages were identified: immature, in early maturation, in advanced maturation, mature, and spawned. Similar findings were reported by Bolognini et al. [10] and Craveiro et al. [27]. Peixoto et al. [13], Campos et al. [15], and Lopes et al. [30] noted the presence of structures known as peripheral bodies or cortical rods during the mature stage, which are characteristic of the penaeid shrimps in the final stage of oocyte maturation. However, in this study, peripheral bodies were not found in *X. kroyeri.*

During all months of sampling, we identified the presence of females in advanced maturation, mature, and spawned stages, indicating that this species engages in reproductive activity throughout the year, which aligns with the findings of the study by Martins et al. [17]. With a predominance of juvenile individuals in April, and of mature females in May and July, months of importance for the life cycle of the population are characterized, in which new individuals arrive for the fishing stock, with a higher presence of females reproducing to continue the cycle of the species in that habitat.

The current closed period established by Interministerial Ordinance No. 47, dated 11 September 2018, from 1 December to 29 February for Espírito Santo, does not include these main periods that protect the breeding and recruitment stock of the species in the studied region. This discrepancy between the closed season established for the area and the reality observed during fishing practices has been reported by fishermen in the municipality of Piúma [31]. It emphasizes the necessity for research to survey the population biology of shrimp, providing identification of times and reproductive size, in order to adjust the legislation and, consequently, conserve this fishing resource and the upper trophic chain that feeds on *X. kroyeri* [7].

This study carried out sampling in all months of the year with the assistance of a member of the fishing community in the municipality of Piúma, so the fishing was carried out in a standardized way according to the local reality. It was observed that the high rate of juvenile individuals appears near the end of the current closed season, indicating it as an effective tool to provide abundant juveniles to enter the population after a break in fishing activity. This ban on fishing is motivated by the species reproducing at least once in the environment and protecting its recruitment to preserve the species. However, the current closed season does not include the species’ main periods of reproduction and recruitment.

The cephalothorax length at which 50% of the population reaches sexual maturity was found to be 1.7 cm in this study, which differs from a study conducted in southeastern Brazil by Castilho et al. [16], where a CL50 value of 1.55 cm was reported for the same species collected in São Paulo. In northeastern Brazil, studies by Reis JR et al. [20] and Silva et al. [21] found values of 1.58 cm and 1.25 cm, respectively. This suggests that the population in this study has a longer cephalothorax first maturation compared to the mentioned regions. In southern Brazil, especially in the state of Santa Catarina, Campos et al. [15] presented in their study a CL50 of 2.4 cm, which is higher than all other data for the same species in the country.

Eutropio et al. [16] and Martins et al. [17] found that the TL50 for *X. kroyeri* in the state of Espírito Santo was 9.02 cm and 6.90 cm, respectively, with the value found in this study being 8.4 cm in total length, which falls within the range observed in the region for half the population to reach sexual maturity.

The sea surface temperature during the sampling period averaged at 24 ± 1.05 °C, unrelated to the number of mature females. Studies by Heckler et al. [32] and Lopes et al. [30] found a positive correlation between an increase in bottom temperature and mature females of *X. kroyeri* on the coasts of São Paulo and Pernambuco, respectively. As for the relationship between rainfall and number of mature females, the correlation was significant in this study, indicating that in the highest rainfall peaks, fewer mature females were present, corroborating studies by Craveiro et al. [27] and Barros et al. [33] for the species *Penaeus schimitti* and *X. kroyeri*, captured in northeastern Brazil, indicating that reproduction occurs in periods of the year with lower rainfall. Temperature and precipitation are motivating factors during many stages in the life cycle of Peneids, influencing growth, the distribution of organisms, and reproductive activity, as well as influencing the availability of nutrients and oxygen sources [34].

In this study, it was found that only rainfall influenced the number of mature females, highlighting that prominent spawning periods occur during the dry season. Recruitment is boosted in the months of increased rainfall, which triggers more nutrients available, creating more favorable survival conditions for young individuals. Another contributing factor is that the turbidity of the water resulting from the rains contributes to the increase in the juvenile rate, proving to be influential in the population structure of the seabob shrimp [33,35].

Brazil has not published fisheries statistics since 2012, leaving the sector without information on how national production is progressing. As a result, it is unclear how much pressure on fish stocks due to fishing activities is presented.

The only study published on the population structure and dynamics of shrimp in Espírito Santo is from 2013, and uses data collected from 2003 to 2004. However, not every month of the year was sampled, raising concerns about the accuracy of the fishing and biological statistical data used in national legislation for Espírito Santo. This highlights the significance of the reproductive and recruitment period presented in this study as information that helps sustainable fishing in the region.

During the closed period, it is essential to develop fishing management strategies that involve fishing communities in testing and adapting the BRDs for local fishing. Medeiros et al. [5] demonstrated the effectiveness of BRDs in trawling for *X. kroyeri* and white shrimp *Penaeus schmitti* in Paraná. These devices not only reduce bycatch but also decreased capture time and improved production quality by yielding larger shrimp with fewer lacerations caused by crabs.

Therefore, government incentives are recommended for updating fishing statistics on the Brazilian coast, testing the implementation of BRDs in the local fishing community, and continuously conducting studies on population dynamics. This will enable a stable understanding of the fishing reality in the region and consequently provide practical information for fisheries legislation.

## 5. Conclusions

Regarding the findings, it is concluded that female seabob shrimp *X. kroyeri* captured on the southern coast of Espírito Santo presented five stages of gonadal development. We found no difference in the sex ratio of male and female individuals, and females are larger and heavier than males in this population. The cephalothorax length CL50 for the first sexual maturity was 1.7 cm, the month of recruitment corresponded to April, and the highest percentage of mature female species was present in May and July, characterizing it as an essential period for the closure of the species in the region studied.

## Figures and Tables

**Figure 1 biology-14-00758-f001:**
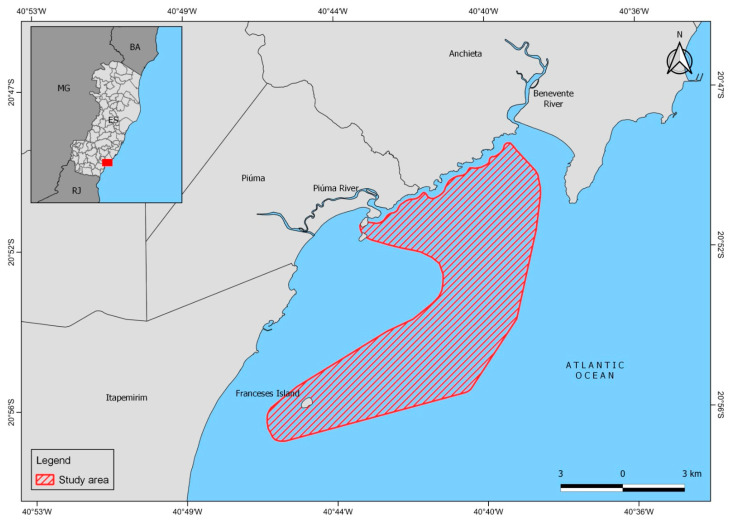
Shrimp, *Xiphopenaeus kroyeri,* collection area on the southern coast of Espírito Santo, Brazil.

**Figure 2 biology-14-00758-f002:**
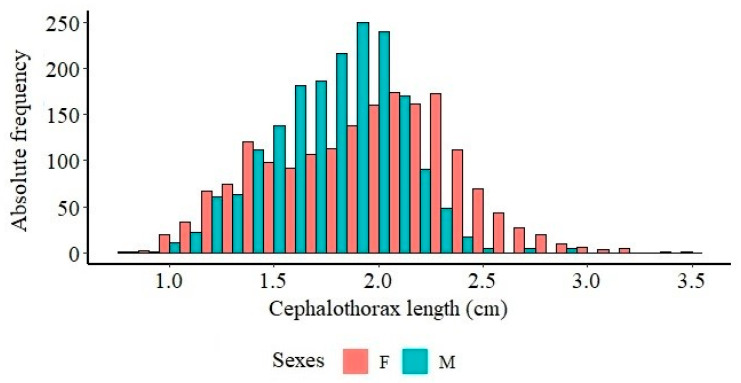
Distribution of absolute frequency by cephalothorax length class of female and male seabob shrimp *Xiphopenaeus kroyeri* captured on the southern coast of Espírito Santo, Brazil. F: Female; M: Male.

**Figure 3 biology-14-00758-f003:**
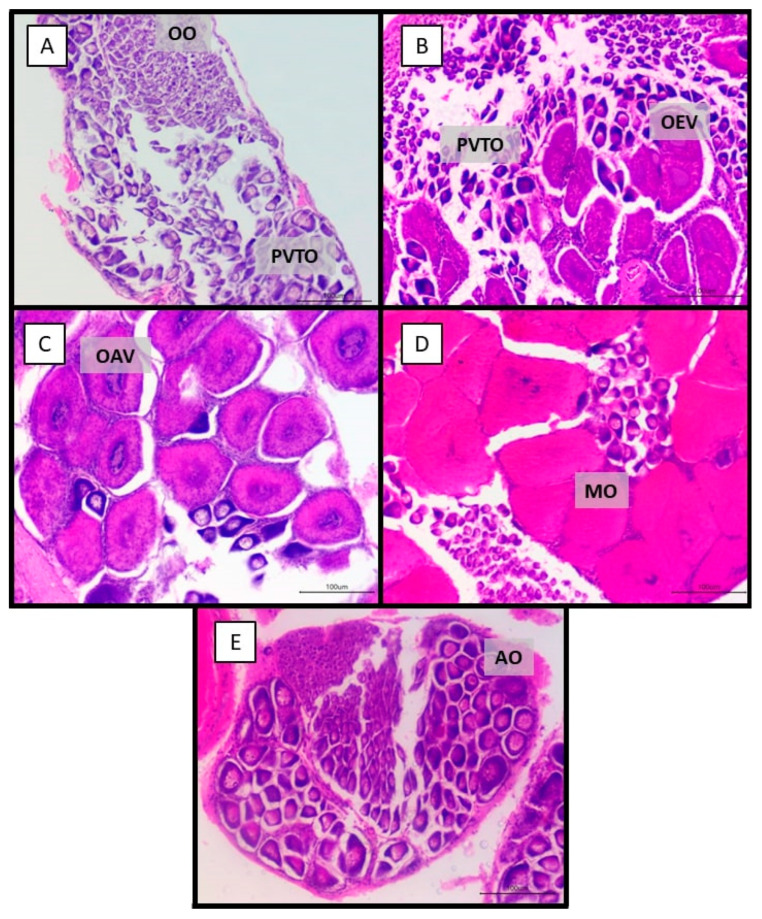
Histological sections characterizing the five stages of gonadal development in female seabob shrimp *Xiphopenaeus kroyeri* captured on the southern coast of Espírito Santo, Brazil. Immature stage (**A**); in early maturation (**B**); stage in advanced maturation (**C**); mature stage (**D**), and spawned stage (**E**). Obj. 20× (OO: oogonia; PVTO: pre-vitellogenesis oocyte; OEV: early vitellogenesis oocyte; OAV: advanced vitellogenesis oocyte; MO: mature oocyte; AO: atretic oocyte).

**Figure 4 biology-14-00758-f004:**
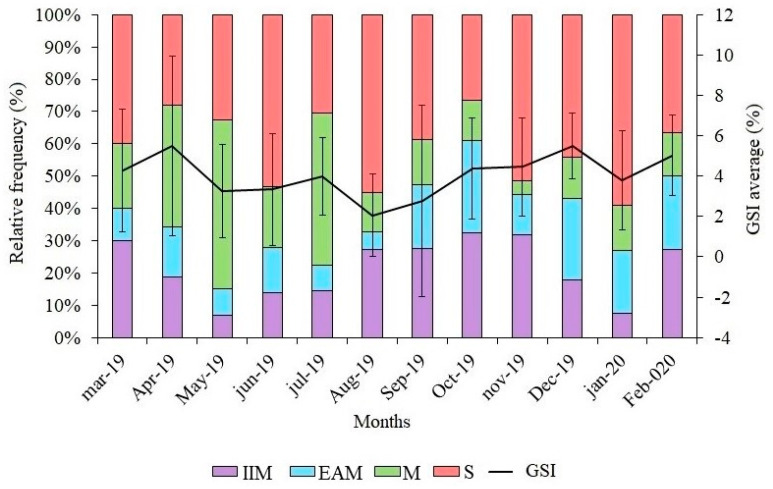
Frequency of ovarian development stages of female seabob shrimp *Xiphopenaeus kroyeri* captured on the southern coast of Espírito Santo, Brazil. IIM: in initial maturation; EAM: in advanced maturation (EAM); M: mature; S: dpawned; GSI: gonadosomatic index.

**Figure 5 biology-14-00758-f005:**
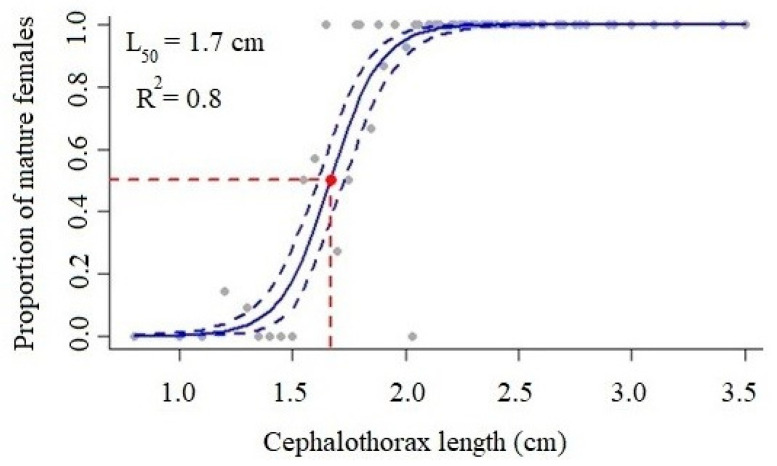
Length at first sexual maturity (CL, cm) of female seabob shrimp *Xiphopenaeus kroyeri*, captured on the southern coast of Espírito Santo, Brazil.

**Figure 6 biology-14-00758-f006:**
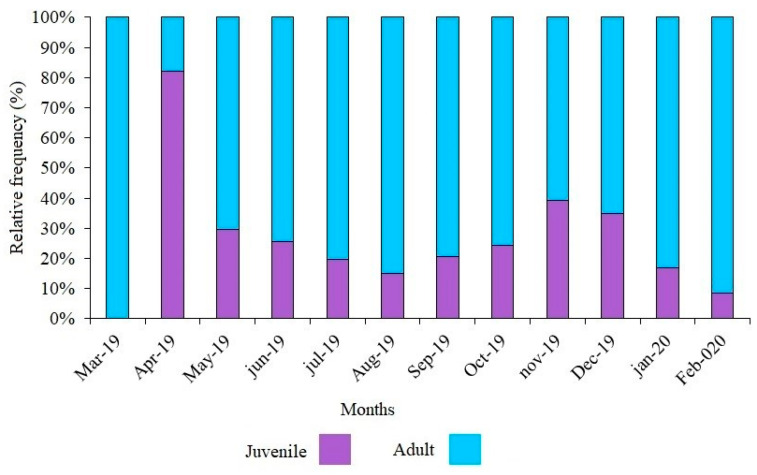
Frequency of young and adult individuals in all sampling months of seabob shrimp *Xiphopenaeus kroyeri*, captured on the southern coast of Espírito Santo, Brazil.

**Figure 7 biology-14-00758-f007:**
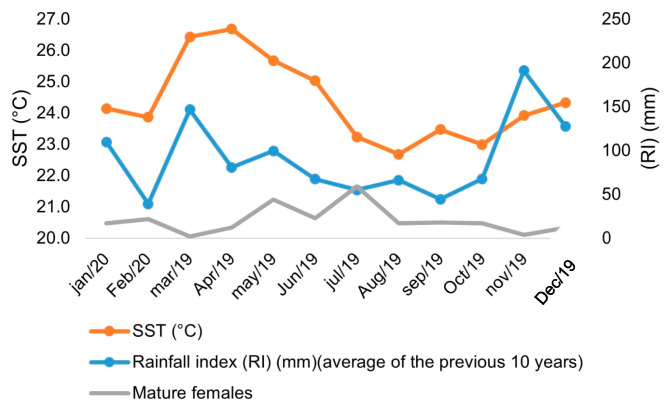
Rainfall index (RI) (mm) (average of the previous 10 years), sea surface temperature (SST) (°C), and ratio of mature females in the months of sampling of seabob shrimp *Xiphopenaeus kroyeri* captured on the southern coast of Espírito Santo, Brazil.

**Table 1 biology-14-00758-t001:** Mean (±SD) of gonad weight and gonadosomatic index (GSI, %) in five maturational stages of seabob shrimp *Xiphopenaeus kroyeri* females captured from March 2019 to February 2020 on the southern coast of Espírito Santo, Brazil.

Parameters	Immature Stage	In Early Maturation	Stage in Advanced Maturation	Mature Stage	Spawned Stage
Gonad weight	0.0001 ± 0.01 ^a^	0.15 ± 0.13 ^b^	0.31 ± 0.18 ^c^	0.37 ± 0.20 ^cd^	0.04 ± 0.05 ^a^
GSI (%)	0.01 ± 0.29 ^a^	2.36 ± 1.59 ^b^	4.04 ± 2.07 ^c^	5.57 ± 1.87 ^cd^	0.86 ± 0.69 ^a^

Different letters in the same column mean statistical differences.

## Data Availability

Data can be requested from the authors upon request.

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
