# Peer review of "Reproductive Dynamics of the Seabob Shrimp Xiphopenaeus kroyeri in Trawl Fisheries in Southeastern Brazil"

_biology, 2025, doi:10.3390/biology14070758_

Round 1
Reviewer 1 Report
Comments and Suggestions for Authors
General Assessment:
The manuscript presents a thorough study of the reproductive biology of Xiphopenaeus kroyeri, with an emphasis on gonadal development, sex ratios, and the potential need for revising the current closed fishing season in Southeastern Brazil. The work is methodologically sound and contributes valuable data to support sustainable fisheries management. However, several areas could benefit from refinement to enhance clarity, scientific rigor, and contextual framing.
Major Comments:
Introduction Section: Lines 45–92: The introduction provides useful background but is somewhat repetitive regarding the ecological and economic importance of the species.
Suggestion: Condense the paragraphs discussing trawling impacts and the biological role of the species to maintain reader engagement. Consider integrating international perspectives for broader relevance.
Materials and Methods: Lines 94–169: While the sampling protocol is well described, the use of statistical methods lacks precision in justification.
Suggestion: Clarify why both parametric and non-parametric tests were selected. Specify the rationale for using the Wilcoxon test and confirm if assumptions for parametric analysis were not met.
Histological Analysis: Lines 136–155: The histological methodology is appropriate, but the selection of 20 specimens monthly appears arbitrary.
Suggestion: Justify the sample size for histological analysis. Consider referencing a power analysis or prior studies that validate the adequacy of this number for statistical robustness.
Results Interpretation: Lines 171–273: The results are presented in detail but could benefit from clearer subsections to distinguish between macroscopic, histological, and statistical outcomes.
Suggestion: Introduce subheadings (e.g., “Macroscopic Classification,” “Histological Validation,” “Statistical Outcomes”) for reader clarity.
Figures and Tables: Figures 2–7 and Table 1: Visual data representations are generally clear but lack explicit reference in the text at some points.
Suggestion: Ensure all figures and tables are explicitly cited and contextualized in the body of the Results section. Improve figure legends to be fully self-explanatory.
Discussion and Management Implications: Lines 274–373: The discussion appropriately challenges current legislation but lacks specific recommendations based on study results.
Suggestion: Offer explicit policy suggestions (e.g., proposed dates for revised closed seasons) derived from the observed peak maturation and recruitment months.
Conclusion: Lines 376–383: The conclusion summarizes findings well but does not emphasize the broader ecological or economic implications.
Suggestion: Expand the conclusion slightly to reflect how findings could influence regional fisheries management or be extrapolated to similar species.
Minor Comments:
Grammar and Style:
Some sentences (e.g., Lines 19–24 and 65–72) are overly complex.
Suggest breaking them into simpler, clearer structures to improve readability.
Terminology Consistency:
Maintain consistency in naming maturation stages (e.g., avoid switching between “IIM” and “In Initial Maturation”).
References:
A few in-text citations lack proper matching with the reference list (e.g., [7] in line 66).
Ensure all references are current, accurate, and correspond with in-text citations.
Author Response
Review 1
General Assessment:
The manuscript presents a thorough study of the reproductive biology of Xiphopenaeus kroyeri, with an emphasis on gonadal development, sex ratios, and the potential need for revising the current closed fishing season in Southeastern Brazil. The work is methodologically sound and contributes valuable data to support sustainable fisheries management. However, several areas could benefit from refinement to enhance clarity, scientific rigor, and contextual framing.
Au: Thank you for your comments.
Major Comments:
Introduction Section: Lines 45–92: The introduction provides useful background but is somewhat repetitive regarding the ecological and economic importance of the species.
Suggestion: Condense the paragraphs discussing trawling impacts and the biological role of the species to maintain reader engagement. Consider integrating international perspectives for broader relevance.
Au: Thank you for your comments. Thank you for your comment. We have rewritten part of the introduction and revised the entire topic.
Materials and Methods: Lines 94–169: While the sampling protocol is well described, the use of statistical methods lacks precision in justification.
Suggestion: Clarify why both parametric and non-parametric tests were selected. Specify the rationale for using the Wilcoxon test and confirm if assumptions for parametric analysis were not met.
Au: Thank you for your comments. The requested information has been included (Line 117-129; 164-168).
Histological Analysis: Lines 136–155: The histological methodology is appropriate, but the selection of 20 specimens monthly appears arbitrary.
Suggestion: Justify the sample size for histological analysis. Consider referencing a power analysis or prior studies that validate the adequacy of this number for statistical robustness.
Au: Thank you for your comments. The requested information has been included (Line 137).
Figures and Tables: Figures 2–7 and Table 1: Visual data representations are generally clear but lack explicit reference in the text at some points.
Suggestion: Ensure all figures and tables are explicitly cited and contextualized in the body of the Results section. Improve figure legends to be fully self-explanatory.
Au: Thank you for your comments. We conducted a review of all figures and tables in the article.
Discussion and Management Implications: Lines 274–373: The discussion appropriately challenges current legislation but lacks specific recommendations based on study results.
Suggestion: Offer explicit policy suggestions (e.g., proposed dates for revised closed seasons) derived from the observed peak maturation and recruitment months.
Au: Thank you for your comments. We conducted a review.
Conclusion: Lines 376–383: The conclusion summarizes findings well but does not emphasize the broader ecological or economic implications.
Suggestion: Expand the conclusion slightly to reflect how findings could influence regional fisheries management or be extrapolated to similar species.
Au: Thank you for your comments. We conducted a review.
Minor Comments:
Grammar and Style:
Some sentences (e.g., Lines 19–24 and 65–72) are overly complex.
Suggest breaking them into simpler, clearer structures to improve readability.
Au: Thank you for your comments. We conducted a review.
Terminology Consistency:
Maintain consistency in naming maturation stages (e.g., avoid switching between “IIM” and “In Initial Maturation”).
Au: Thank you for your comments. We conducted a review.
References:
A few in-text citations lack proper matching with the reference list (e.g., [7] in line 66).
Ensure all references are current, accurate, and correspond with in-text citations.
Au: Thank you for your comments. We conducted a review.
Reviewer 2 Report
Comments and Suggestions for Authors
Dear authors,
Please see the attached comments in the PDF file
Best regards,

Author Response
Review 2
Several suggestions for corrections were made in the Word file.
Au: Thank you for your contribution. I would like to inform you that all suggestions were accepted and are highlighted in the file in yellow.
Reviewer 3 Report
Comments and Suggestions for Authors
Review for the paper “Reproductive dynamics of the seabob shrimp Xiphopenaeus kroyeri in trawl fisheries in Southeastern Brazil” by Amanda Soares dos Santos and co-authors submitted to “Biology”.
The authors of this research paper conducted an analysis of the gonadal development stages of Xiphopenaeus kroyeri, a species experiencing significant capture pressure in the western Atlantic. Their study focused on the relationships between climatic factors and the reproductive biology of this shrimp in the Southeastern Atlantic Ocean of Brazil. The authors found that the female gonads could be categorized into distinct stages of maturation, ranging from immature to spawned. The recruitment period, characterized by the emergence of juvenile individuals, was primarily noted in April. Temperature and precipitation data were analyzed, revealing a negative correlation between rainfall and the number of mature females present.
The results of this study may have important implications for fisheries management, particularly in the context of setting appropriate closed seasons for X. kroyeri.
Recommendations.
Abstract.
L 24-26. This sentence is unclear. The authors should rephrase it.
L 40-41. Consider replacing “The results suggest a review of the current closed season established for the species” with “The results suggest that the current closed season for the species should be reconsidered”.
Introduction.
L 49. Could the authors provide landing data for Xiphopenaeus kroyeri in Brazil?
L 61. The authors should report the specific criteria that define the closed period for X. kroyeri in the study area. What adaptations or changes in fishing practices have been made in Espirito Santo to reduce the ecological damage caused by trawling?
L 85. The text should include more details about previous studies.
Materials and Methods.
Figure 1. The font size of the coordinate grid should be increased.
L 163-164. The authors should explain the resolution and timeline of the environmental data collection to clarify how accurately these factors were correlated with reproductive patterns.
Results.
L 178-183. The authors should include the minimum and maximum levels for all average length measurements.
L 194-197. It would be useful to compare the size-frequency distributions of males and females using a chi-square test.
L 206-211. The authors should include minimum and maximum levels for the average oocyte diameter measurements.
Figure 3. The scale bar is not clearly visible. The authors should either increase the font size or include the scale bar information in the figure caption.
Table 1: The authors should explain the different superscript letters in the table. In the second row, "IGS" should be replaced by "GSI."
Figure 7: "dez/19" should be replaced by "Dec/19".
Discussion.
L 297. Peripheral bodies are typical for penaeid species, so their absence requires further explanation.
L 313. The authors should recommend the best fishing season for this species in this region.
L 323-334. Why is there regional variability in the cephalothorax length (CL50) and total length (TL50) at first maturity across different states? The authors should clarify whether this is due to environmental differences, fishing pressure, genetic adaptations, or sampling methodologies.
L 346. Why was no correlation found between the sea surface temperature and the number of mature females in this study, despite previous demonstrating such a relationship?
Author Response
Review 3
The authors of this research paper conducted an analysis of the gonadal development stages of Xiphopenaeus kroyeri, a species experiencing significant capture pressure in the western Atlantic. Their study focused on the relationships between climatic factors and the reproductive biology of this shrimp in the Southeastern Atlantic Ocean of Brazil. The authors found that the female gonads could be categorized into distinct stages of maturation, ranging from immature to spawned. The recruitment period, characterized by the emergence of juvenile individuals, was primarily noted in April. Temperature and precipitation data were analyzed, revealing a negative correlation between rainfall and the number of mature females present.
The results of this study may have important implications for fisheries management, particularly in the context of setting appropriate closed seasons for X. kroyeri.
Au: Thank you for your contribution.
Recommendations.
Abstract.
L 24-26. This sentence is unclear. The authors should rephrase it.
Au: Thank you for your contribution. The sentence was rewritten (Line 25-26).
L 40-41. Consider replacing “The results suggest a review of the current closed season established for the species” with “The results suggest that the current closed season for the species should be reconsidered”.
Au: Thank you for your contribution. We replaced the sentence as recommended (Line 40-41).
Introduction.
L 49. Could the authors provide landing data for Xiphopenaeus kroyeri in Brazil?
Au: Thank you for your contribution. Data on landings of this species are non-existent.
Materials and Methods.
Figure 1. The font size of the coordinate grid should be increased.
Au: Thank you for your contribution. The figure has been enlarged.
Results.
L 178-183. The authors should include the minimum and maximum levels for all average length measurements.
Au: Thank you for your contribution.The information has been included (line 175-178)
L 206-211. The authors should include minimum and maximum levels for the average oocyte diameter measurements.
Au: Thank you for your contribution.The information has been included (line 203-209)
Figure 3. The scale bar is not clearly visible. The authors should either increase the font size or include the scale bar information in the figure caption.
Au: Thank you for your contribution. The figure has been reworked.
Table 1: The authors should explain the different superscript letters in the table. In the second row, "IGS" should be replaced by "GSI."
Au: Thank you for your contribution. The table has been revised.
Figure 7: "dez/19" should be replaced by "Dec/19".
Au: Thank you for your contribution. Figure 7 has been revised.